EMBO
reports

# Metabolic energy sensing by mammalian CLC anion/proton exchangers

Matthias Grieschat[1,†], Raul E Guzman[2,†], Katharina Langschwager[1,†], Christoph Fahlke[2] (ID) & Alexi K Alekov[1,*] (ID)

## Abstract

CLC anion/proton exchangers control the pH and [Cl⁻] of the endolysosomal system that is essential for cellular nutrient uptake. Here, we use heterologous expression and whole-cell electrophysiology to investigate the regulation of the CLC isoforms ClC-3, ClC-4, and ClC-5 by the adenylic system components ATP, ADP, and AMP. Our results show that cytosolic ATP and ADP but not AMP and $Mg^{2+}$-free ADP enhance CLC ion transport. Biophysical analysis reveals that adenine nucleotides alter the ratio between CLC ion transport and CLC gating charge and shift the CLC voltage-dependent activation. The latter effect is suppressed by blocking the intracellular entrance of the proton transport pathway. We suggest, therefore, that adenine nucleotides regulate the internal proton delivery into the CLC transporter machinery and alter the probability of CLC transporters to undergo silent non-transporting cycles. Our findings suggest that the CBS domains in mammalian CLC transporters serve as energy sensors that regulate vesicular $Cl^-/H^+$ exchange by detecting changes in the cytosolic ATP/ADP/AMP equilibrium. Such sensing mechanism links the endolysosomal activity to the cellular metabolic state.

**Keywords** adenine nucleotide regulation; CBS domain; CLC transporter
**Subject Categories** Membranes & Trafficking; Metabolism

## Introduction

CLC anion channels and anion/proton exchangers fulfill indispensable functions in nerve and muscle excitation, endocytosis, exocytosis, and lysosomal function [1,2]. The mammalian isoforms ClC-3 to ClC-7 are localized mainly on intracellular vesicles and regulate the vesicular acidity and $Cl^-$ content [2]. CLC proteins assemble as dimers of two identical subunits with parallel orientation and separate ion transport pathways. The intracellular carboxy terminus (C-terminus) of each subunit contains a so-called Bateman domain with two distinct cystathionine β-synthase domains, CBS1 and CBS2 [3–7]. CBS domains are adenine nucleotide-binding structures found in many unrelated protein families; their physiological function has been established by the association with various human hereditary diseases [8–11]. A good example is the AMPK protein family (AMP-activated protein kinases) in which the CBS domains serve as energy sensors detecting changes in the ATP/ADP/AMP levels (see [12] for a review).

Currently, only the structure of the ClC-5 C-terminus has been solved featuring an adenine nucleotide-binding pocket formed between the two CBS domains with attached either one molecule ATP or ADP ([6], Fig 1A, Movies EV1 and EV2). To our knowledge, the AMP-occupied and the nucleotide-free CBS domain structures have not been characterized yet for any of the mammalian CLC transporters. Nevertheless, large-scale ATP-induced conformational changes of the ClC-5 C-terminus demonstrate the regulatory capacity of the nucleotide-binding pocket [13]. Combined, the structural data suggest that mammalian CLC transporters are able to detect and respond to changes in the cellular energy status; however, the functional effects of nucleotide binding are insufficiently understood. Controversial findings have been reported on the very similar isoforms, ClC-4 and ClC-5 [14,15]. Whereas ClC-4 was found to discriminate between ATP and ADP [14], ClC-5 was reported to behave similarly in the presence of ATP, ADP, and AMP [15]. Of note, the release or production of metabolic energy results predominantly in ATP/ADP/AMP interconversion which does not change the total cellular content of these nucleotides [12]. Hence, the different effects of adenine nucleotides on ClC-4 suggest that the CLC CBS domains may serve as energy sensors [14]. In contrast, the identical functional effects of ATP, ADP, and AMP on ClC-5 implicate that nucleotide binding in mammalian CLC transporters is physiologically irrelevant [15].

To clarify whether mammalian CLC transporters can sense and respond to metabolic energy changes, we reinvestigated the effects of ATP, ADP, and AMP on ClC-4 and ClC-5. In addition, we tested whether adenine nucleotides can functionally regulate the neuronal transporter ClC-3. Our data showed that all investigated CLC isoforms can discriminate between cytosolic ATP and AMP and are, therefore, able to detect changes in the cellular energy status. Gating

---

1 Institute of Neurophysiology, Hannover Medical School, Hannover, Germany
2 Institute of Complex Systems, Zelluläre Biophysik (ICS-4), Forschungszentrum Jülich, Jülich, Germany
*Corresponding author. Tel: +49 511 532 5543; Fax: +49 511 532 9391; E-mail: alexi.alekov@gmail.com
†These authors contributed equally to this work

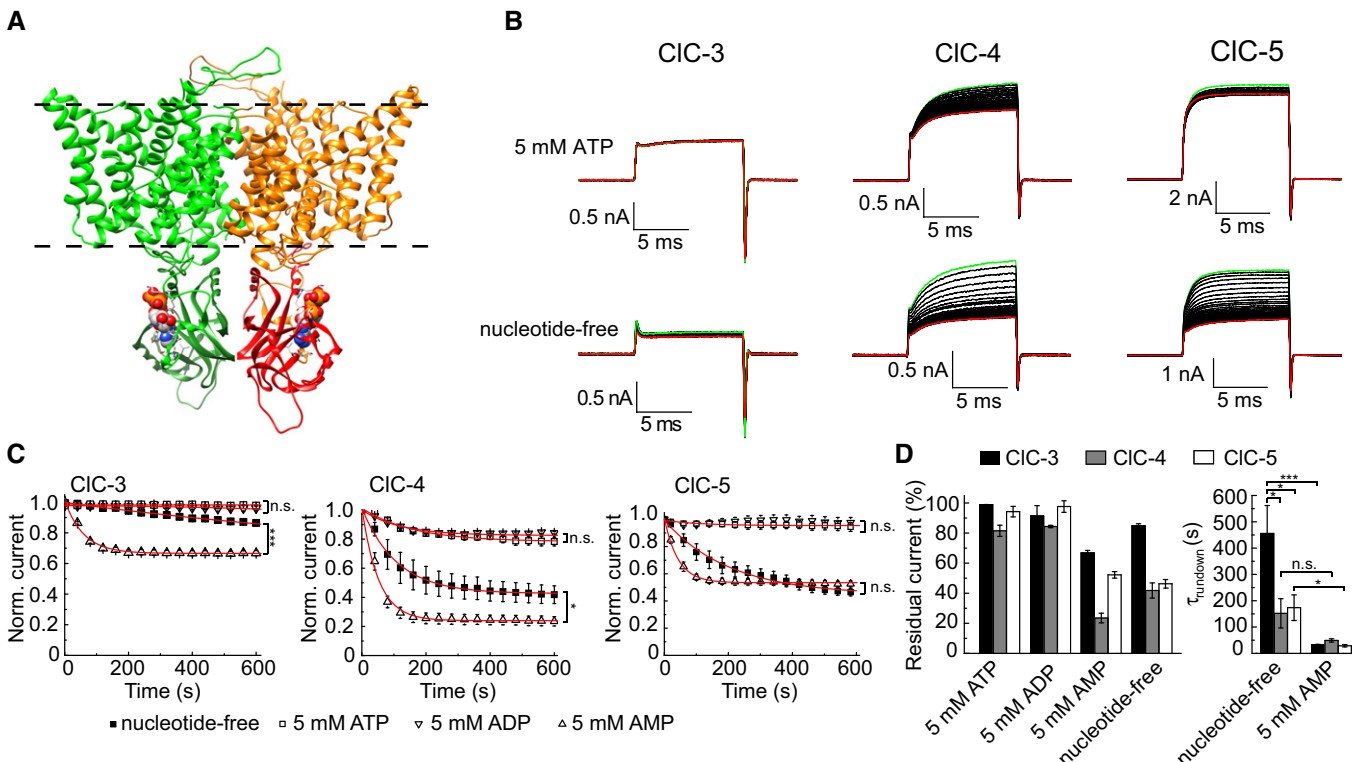

**Figure 1. Cytosolic adenine nucleotides regulate the function of ClC-3, ClC-4, and ClC-5.**

A  Homology model depicting the ClC-5 dimeric structure with bound ATP molecules represented as spheres. Nucleotide-coordinating residues in the CBS domains are represented as sticks. Dashed lines indicate the hydrocarbon boundaries of the lipid bilayer.

B  Representative leak-subtracted ClC-3, ClC-4, and ClC-5 whole-cell currents measured with (upper row) or without (lower row) 5 mM ATP added to the pipette solution. Currents were elicited every 10 s by voltage steps from 0 mV to +140 mV, followed by a step to -100 mV to maximize the off-gating currents visible as sharp peaks at the end of the pulses. The recording started shortly (10s–30s) after obtaining a whole-cell configuration. The first and last recorded currents in a series are depicted in green and red, respectively. NMDG-based solutions were used to characterize ClC-4.

C  Effects of various adenine nucleotides or their washout on the ionic transport of ClC-3, ClC-4, and ClC-5. The current amplitudes were measured at the end of the test pulses from experiments as shown in (B) and normalized to the initial current amplitudes obtained after establishing the whole-cell configuration. Some of the experimental data points have been omitted for clarity. Monoexponential fits to the data are depicted as red lines. Asterisks indicate statistical significances evaluated by two-sample $t$-test analysis of the residual currents determined at the end of each 10-min measurement (*$P < 0.05$; ***$P < 0.001$, n.s., not significant; $n$ = 5–11).

D  Left panel: residual currents determined at the end of 10-min measurements as shown in (C). Right panel: time constants ($\tau$) of the current rundown obtained by fitting monoexponential functions to individual measurements as depicted in (C), ($n$ = 5–11). Asterisks indicate statistical significances evaluated using two-sample $t$-test analysis (*$P < 0.05$; ***$P < 0.001$, n.s., not significant).

Data information: Error bars in all panels represent SEM.

current measurements revealed that adenine nucleotides alter the CLC voltage-dependent activation. Mutagenesis and further biophysical analysis suggested that the effects are coupled to the CLC proton transport machinery.

## Results and Discussion

### The ion transport of ClC-3, ClC-4, and ClC-5 is differentially regulated by intracellular ATP, ADP, and AMP

We transfected HEK293T cells and used whole-cell electrophysiology to investigate the functional effects of adenine nucleotides on three CLC anion/proton exchanger isoforms, the human ClC-4 and ClC-5, and the membrane-localized mouse splice variant ClC-3c [16]. The expression resulted in large macroscopic currents with the

specific biophysical properties of the corresponding isoforms (Fig 1B) [17]. In whole-cell mode, the content of the cytoplasm is exchanged for the pipette solution by diffusion [18]. Hence, tracking the CLC current amplitude after establishing whole-cell configuration can be used to detect effects resulting from the washout of endogenously present cytosolic nucleotides or from their substitution by nucleotides added to the pipette solution. To mimic the physiological conditions, 5 mM $MgCl_2$ were added to pipette solution (unless explicitly stated otherwise).

Experiments with 5 mM ATP in the intracellular solution revealed a minor CLC transport decrease (Fig 1B–D). As previously demonstrated [14,15], a much larger CLC transport decrease was observed for all the investigated isoforms in experiments when nucleotide-free pipette solution was used (Fig 1B–D). Analyzing the data with and without applied leak subtraction excluded artifacts due to slow changes of the seal resistance during the experiments

(Appendix Fig S1). Please note that also for ClC-3, the steady state was not reached when using nucleotide-free solution; therefore, these measurements may underestimate the effects of ATP in this isoform.

The above results suggest that intracellular ATP binding enhances the ion transport of ClC-3, ClC-4, and ClC-5. To support this conclusion, we tested the reversibility of the effects using excised inside-out patches. The intracellular nature of the investigated transporters resulted in small current amplitudes and precluded measurements with standard Cl$^-$-based solutions. To overcome this limitation, we used external SCN$^-$-based solution (added in the pipette in the inside-out configuration) that partially uncouples CLC transport and increases the ClC-5 current amplitudes [19]. The effects are, therefore, not directly comparable to effects quantified using Cl$^-$-based solutions. ATP application to the internal side of excised patches reversibly increased the CLC current amplitude (Appendix Fig S2). Therefore, the current rundown induced by the washout of cytoplasmic ATP does not result from the internalization of CLC proteins localized in the plasma membrane. Low current amplitudes precluded analogous measurements with ClC-3 and ClC-4 and with ADP and AMP.

Next, we mutated one of the amino acids coordinating the ATP molecule in the ClC-5 nucleotide-binding pocket (mutation Y617A, [6]). As expected, and as previously reported [15], the mutant abolished the current decrease associated with cytosolic ATP washout (Appendix Fig S3A and B). Similar results were obtained for the analogous mutation in ClC-4 (Appendix Fig S3C). These data demonstrate the role of the C-terminal CBS domains in the adenine nucleotide regulation of mammalian CLC transport.

To test whether mammalian CLC transporters can detect metabolic energy changes, we performed experiments with intracellular ADP and AMP. The ionic transport of the investigated isoforms was not substantially reduced when 5 mM ADP was added to the pipette solution. In contrast, a large reduction was observed with 5 mM AMP in the pipette (Fig 1B). Analyzing the residual currents reached at the end of the 10-min measurements (the steady-state amplitudes) revealed that their levels differ between experiments with AMP-containing pipette solution and nucleotide-free measurements (Fig 1C and D). This difference suggests that AMP can displace ATP from the C-terminal CLC-binding pocket. To test this hypothesis, we conducted experiments with both 1 mM AMP and 5 mM ATP added to the pipette solution. In agreement with the proposed competitive displacement of ATP by AMP (see also [13,20]), the ion transport decrease in these experiments was larger than the decrease in experiments conducted with only ATP in the pipette (Appendix Fig S4). Kinetic analysis revealed that the AMP-induced rundown in ClC-3 and ClC-5 was faster compared to nucleotide-free measurements (Fig 1C and D). Comparing the whole-cell capacitances and series resistances of experiments with different nucleotides indicated that the effects are not due to differences in the cell sizes or in the patch pipette diameters (Appendix Fig S1). However, as whole-cell measurements do not allow precise control over the perfusion speed, the significance of the encountered different kinetics needs to be demonstrated.

The here-presented findings are in line with previous investigations into the interaction between adenine nucleotides and mammalian CLC transporters [14,20]. They are also coherent with the different effects of AMP and ATP on the *Arabidopsis* AtCLC

transporter [21]. Our data are also in agreement with most of the findings of a previous comprehensive investigation on ClC-5, except for the published ion current increase observed in experiments with intracellular AMP (see [15]). Further experiments are required to uncover the reason for the discrepancy. It should be reiterated, however, that the investigated ClC-3, ClC-4, and ClC-4 transporters are intracellular proteins which limit the current amplitudes measured in the excised patch configuration. Similarly, the effects analyzed in the aforementioned ClC-5 study [15] were very small. Therefore, it is possible that endogenous conductances as reported for the *Xenopus laevis* expression system [22,23], probably activated by AMP and amplified by applied leak subtraction procedures, might have obscured the CLC current reduction induced by this nucleotide.

In summary, our experiments reveal that cytosolic ATP and ADP increase the ion transport rates of ClC-3, ClC-4, and ClC-5. In contrast, cytosolic AMP decreased CLC ion transport. The different magnitudes of the effects in ClC-3, ClC-4, and ClC-5 suggest that adenine nucleotide regulation might be optimized to match the physiological roles of the specific CLC isoforms.

### Cytosolic adenine nucleotides regulate the CLC transport cycle and voltage-dependent gating

Cytosolic adenine nucleotides regulate the ion currents of the CLC channels ClC-1 and ClC-2 by altering their voltage-dependent activation [24–26]. Similar to CLC channels, the here-investigated CLC transporters exhibit pronounced voltage dependence [17,27,28]. Therefore, we used gating current analysis [17,19,27] to test whether adenine nucleotides alter the voltage-dependent activation of ClC-5. Specifically, we calculated the gating charge using the area beneath the gating currents of this transporter (Fig 2A). The analysis revealed that ATP, ADP, and AMP all shift the ClC-5 activation toward more positive voltages (Fig 2B, Appendix Fig S5A, Appendix Table S1). This shift may contribute to the AMP-induced current decrease; however, it cannot explain the ClC-5 transport increase observed with ATP and ADP in the pipette solution.

It was previously demonstrated that impaired delivery of internal protons into the CLC transport machinery reduces CLC ion transport but increases the CLC gating currents [17,19,27]. Investigations by us [17,19] indicate that this behavior might reflect entry into a silent non-transporting mode and that the ratio between the gating charge and the ion current at one fixed voltage can be used to estimate the number of silent non-transporting CLC proteins (see Fig 2A). Applying this analysis to ClC-5 revealed that ATP and ADP both decrease the aforementioned ratio (Fig 2A and B). We suggest, therefore, that these nucleotides enhance CLC ion transport by reducing the probability that ClC-5 undergoes silent non-transporting cycles. As AMP binding enlarges the aforementioned ratio (Fig 2A and B), our hypothesis implies that this nucleotide increases the silent transport cycle probability. Of note, the apparent ClC-5 gating charge in experiments with 5 mM ATP differed from the one measured with both 5 mM ATP and 1 mM AMP added to the pipette solution (Appendix Fig S5B). This difference provides additional evidence that AMP can displace ATP from the CBS nucleotide-binding pocket.

Analyzing the gating currents of ClC-3 revealed that cytosolic adenine nucleotides also affect both the voltage dependence and the ratio between the ClC-3 gating charge and ion currents (Fig 2C).

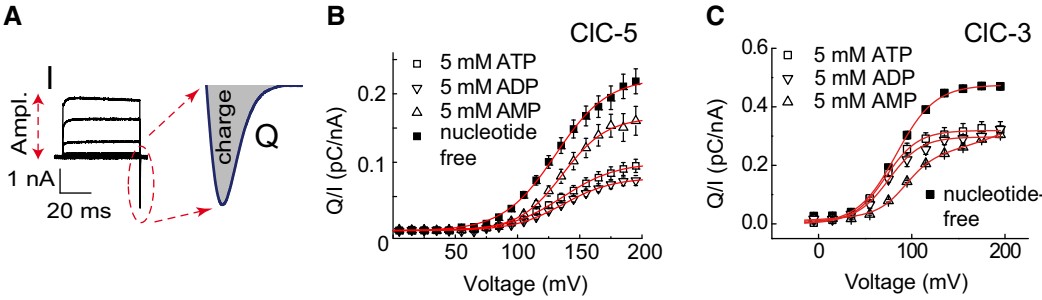

**Figure 2.** Adenine nucleotides regulate the CLC voltage dependence and relative gating charge amplitudes.

A  Schematic representation of the gating charge analysis. The gating charge Q was obtained by calculating the surface beneath the off-gating currents (enlarged in gray and denoted by "charge"). Relative gating charge amplitudes in (B) and (C) were obtained by dividing the gating charge Q by the ion current amplitude I ("Ampl.") at +165 mV.

B  Voltage dependence of the WT ClC-5 off-gating charge normalized to the ionic current at +165 mV in the absence or presence of adenine nucleotides ($n$ = 7–12). Error bars represent SEM. Red lines depict Boltzmann fits to the data; fit parameters are provided in Appendix Table S1.

C  Voltage dependence of the WT ClC-3 off-gating charge normalized to the ionic current at +165 mV in the absence or presence of adenine nucleotides ($n$ = 4–5). Error bars represent SEM. Red lines represent Boltzmann fits to the data; fit parameters are provided in Appendix Table S2.

However, in contrast to ClC-5, cytosolic AMP induced a stronger right shift of the ClC-3 voltage-dependent activation. This explains well the different responses of ClC-3 and ClC-5 to AMP when compared to measurements with nucleotide-free pipette solution (Fig 1) and hints at the physiological specialization of these CLC isoforms.

### The proton transport pathway is important for CLC adenine nucleotide regulation

Voltage-dependent gating in CLC channels relies upon a conserved negatively charged glutamate residue at the extracellular entrance of the anion transport pathway, the so-called gating glutamate $Glu_{ext}$ [4]. $Glu_{ext}$ plays also a critical role in the voltage-dependent gating of ClC-3, ClC-4, and ClC-5 [27,29–32]. This led us to evaluate the specific role of this residue in the adenine nucleotide regulation of mammalian CLC transporters. To this end, we characterized the effect of cytosolic ATP on the pathogenic mutant E211G that neutralizes the electric charge of the ClC-5 $Glu_{ext}$ E211 [33]. Experiments with nucleotide-free pipette solution revealed that ATP depletion reduces the ion transport of E211G ClC-5 (Fig 3A and B, Appendix Fig S6). Thus, the $Glu_{ext}$-dependent activation process is not the sole determinant of the CLC adenine nucleotide regulation.

CLC channels, while unable to transport protons, have preserved parts of the proton transport machinery of the archetypic CLC Cl⁻/H⁺ exchangers [34]. In this regard, it is established that $Glu_{ext}$ forms the external entrance of the CLC proton transport pathway in CLC anion/proton exchangers [29,30,35]. The here-investigated ClC-5 E211G also abolishes proton transport and converts ClC-5 into a passive Cl⁻ conductor resembling the classical CLC channels [33]. Therefore, the strong cytosolic (internal) pH dependence of the ATP regulation in ClC-1 channels [36,37] and the preserved ATP action in the ClC-5 transporter $Glu_{ext}$ mutant E211G (Fig 3) both hint at the existence of common molecular mechanisms underlying adenine nucleotide regulation in CLC channels and transporters.

To provide support for the above hypothesis, we considered that the chloride and proton transport pathways in CLC Cl⁻/H⁺ exchangers partially overlap only at their extracellular part

containing the gating glutamate $Glu_{ext}$ [4,35,38,39] but diverge toward the intracellular protein side. Therefore, we investigated mutation E268Q that specifically blocks the intracellular proton access in ClC-5 [19,27]. The non-transporting phenotype of the mutant [19,27] prevents ion current measurements; however, gating charge analysis revealed that the nucleotide effects on ClC-5 voltage dependence are suppressed (Appendix Fig S7, Appendix Table S3). Hence, adenine nucleotide regulation of voltage-dependent gating depends on the internal (cytosolic) delivery of protons into the CLC transporter machinery. This conclusion is supported by the lack of ATP effects in experiments with lower intracellular pH [15] and by the abolished internal pH dependence of mutant ClC-5 E268Q [19,27]. Finally, the shifts in the voltage-dependent activation induced by adenine nucleotides (Fig 2) resemble effects induced by lower cytosolic pH [19,27] and may be similarly explained by an increased delivery of internal protons into the CLC protein. Of note, the intracellular entrance of the CLC proton transport pathway is localized in the proximity of the region connecting the C-terminus and the CLC transmembrane core [19,39–41]. Therefore, the recently demonstrated long-range communication between the C-terminus and the CLC transmembrane core of the anion channel homolog CLH-3b [42] also hints at nucleotide-induced rearrangements that are likely to be propagated toward the transmembrane core and to affect the injection of protons into the CLC transporting machinery.

### Nucleotide chemistry and cytosolic Mg²⁺ are important for the adenine nucleotide regulation of CLC transport

We next performed experiments to investigate the specificity of CLC adenine nucleotide regulation. It was previously reported that the non-hydrolysable ATP analog adenylyl imidodiphosphate (AMP-PNP) cannot augment ClC-4 transport currents [14]. Patch-clamp measurements with 5 mM AMP-PNP in the pipette solution revealed that ClC-5 behaves similarly (Fig 4A–C). The comparable ClC-5 current rundowns observed in the presence of intracellular AMP-PNP and in the presence of AMP (see Fig 1) suggest that the terminal bridge oxygen of the ATP triphosphate chain is important for the

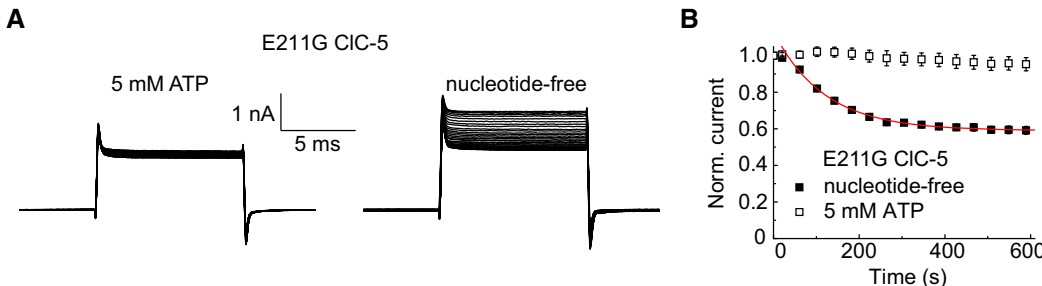

**Figure 3. Mutating the Glu$_{ext}$ E211 does not abolish ATP effects.**

A Representative whole-cell recordings of cells expressing E211G ClC-5 upon voltage pulses to +140 mV from a −60-mV holding potential applied in 10-s intervals starting shortly after obtaining the whole-cell configuration with (left) or without (right) 5 mM ATP in the pipette solution. Leak subtraction was not applied. Asymmetric [Cl⁻] was used (110 mM internal NaCl substituted by TrisSO$_4$) to achieve a stable negative reversal potential of < −50 mV allowing to recognize recordings with high unspecific leak current contamination.

B Normalized time course of the steady-state E211G ClC-5 current amplitudes measured with ($n = 5$) or without ($n = 4$) internal ATP. Error bars show SEM. Red line represents a monoexponential fit to the data with a time constant of $108 \pm 8$ s (SEM); currents measured with ATP-free pipette solution declined to $57.9 \pm 2.6\%$ (SEM) of the initial amplitude.

regulation of mammalian CLC transport. In the same set of experiments, we investigated the effects of S-adenosyl-methionine (SAM or AdoMet), a small molecule that activates human cystathionine-beta-synthases by interacting with their CBS domains [43,44]. In contrast to AMP-PNP, the rundown observed with SAM in the pipette solution resembled nucleotide-free measurements (Fig 4A and B). Therefore, a phosphate moiety seems to be required for the adenine nucleotide regulation of CLC transport, either to stabilize nucleotide binding or to propagate conformational changes induced by such binding.

In mammalian cells, adenine nucleotides form biologically active complexes with Mg$^{2+}$ [45]. To our knowledge, it has not been investigated whether Mg$^{2+}$ ions play a role in the regulation of CLC transport. However, a reduced [Mg$^{2+}$] was used in the previous study [14], in which ADP was found unable to enhance ClC-4 transport. This led us to eliminate the MgCl$_2$ from our pipette solution. In contrast to experiments with 5 mM MgCl$_2$ in the pipette (see Fig 1), a significant ClC-5 current rundown was observed after obtaining

the whole-cell configuration (Fig 4A and B). These experiments demonstrate that Mg$^{2+}$ ions are important for the adenine nucleotide regulation of mammalian CLC transport. It has been previously shown that the ClC-5 CBS domains can bind ADP also in the absence of Mg$^{2+}$ [6]. As such binding does not increase CLC ion transport (Fig 4A), it is likely that Mg$^{2+}$ ions fine-tune the conformational changes that lead to full transport activation.

At this point, we can only speculate about the physiological role of the here-described adenine nucleotide regulation. However, the pivotal role of intracellular CLC transporters in endosomal and lysosomal function renders such regulation physiologically meaningful and creates a link between the cellular metabolic state and the activity of the endolysosomal system. Therefore, the effects of pathogenic mutations associated with Dent disease on both the voltage dependence and relative gating charge amplitudes of ClC-5 [40] suggest that nucleotide sensing might be important for the observed renal pathophysiology in the affected patients. As vesicular secondary active exchangers [17,29,30], ClC-3, ClC-4, and ClC-5

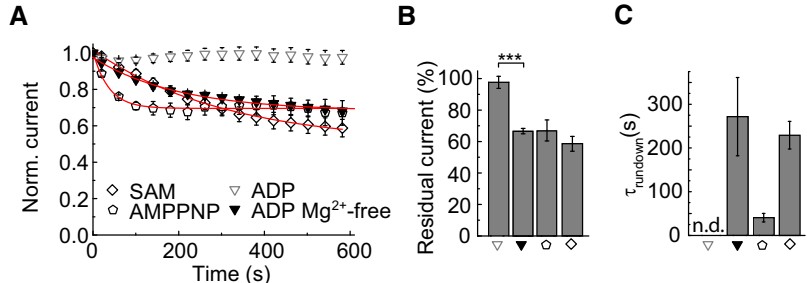

**Figure 4. Nucleotide chemistry and Mg$^{2+}$ are important for the regulation of CLC transport.**

A Effects of different adenine nucleotides or their washout on the ClC-5 current amplitudes ($n = 4–7$). The data were normalized to the initial current amplitude determined shortly after establishing whole-cell configuration. Some of the data points are omitted for clarity. For comparison, the ADP data set from Fig 1 is also depicted. Red lines indicate monoexponential fits to the averaged data.

B Averaged residual currents determined after 600-s cell dialysis as shown in (A) and normalized to the current at the start of the measurements ($n = 4–7$). Asterisks indicate statistically significant differences from a two-sample $t$-test analysis with ***$P < 0.001$.

C Averaged rundown time constants obtained from fits with monoexponential decay functions as shown in (A). n.d.—not defined ($n = 4–7$).

Data information: Error bars in all panels represent SEM.

utilize the electrochemical gradient built up by ATP-dependent pumps [2]. Hence, an [AMP] increase upon depletion of the cellular chemical energy sources will suppress coupled CLC transport and reduce ATP consumption. In addition, the associated larger gating charges will increase the vesicular membrane capacitance [17,19] and reduce the energy barrier opposing the V-ATPase-mediated vesicular acidification [17]. In this regard, the selective apoptosis of hippocampal neurons in ClC-3 knockout mice [46] might reflect reduced adaptability and higher susceptibility to cell death induced by energy deprivation. ClC-3 sensing of the ATP/ADP/AMP ratio might control insulin production by defining the rate of granular acidification [47]. Analogously, pathogenic ClC-4 loss-of-function mutations [48,49] might reduce the ability of neurons to react to metabolic stress and accordingly might affect cognitive function. Drawing an analogy to the AMPK family proteins [12], our findings suggest that mammalian CLC exchangers might participate in a survival program activated during cell starvation or other extreme conditions. Specifically, the expression pattern and vesicular localization of the here-investigated isoforms (recently reviewed in [2]) suggest that the metabolic energy dependence of intracellular CLC transport might be highly relevant during various ischemic, hypoxic, or anoxic conditions of the kidney, heart, and brain, including acute injuries, diabetes, heart attack, and stroke [50–56].

# Materials and Methods

### Cell culture, construct generation, expression

DNA sequences encoding for the investigated isoforms ClC-3, ClC-4, and ClC-5 were inserted into specially functionalized expression vectors and controlled by sequencing. For ClC-4, the bicistronic pRcCMV vector contained additionally the CD8 surface antigen, translationally controlled by an internal ribosome entry site (IRES [57]) to allow identification of successfully transfected cells with anti-CD8 antibody-coated polystyrene beads (Dynabeads CD8; Gibco, Life Technologies, Darmstadt, Germany). For the expression of WT ClC-5, the DNA sequences of the RFP variant mCherry and the CLC protein were fused together into the pRcCMV expression vector (described elsewhere [19]), allowing the selection of transfected cells by fluorescence. The used ClC-3c eGFP or mRFP constructs into the FsY1.1G.W. or p156rrL vectors are also described elsewhere [16]. Point mutations for amino acid substitutions or truncations were generated using the QuikChange site-directed mutagenesis kit (Stratagene, La Jolla, CA, USA).

HEK293T cells were transiently transfected with 5–10 μg plasmid DNA using calcium phosphate precipitation [58]. Cells were cultured in DMEM (Gibco, Life Technologies, Darmstadt, Germany), supplemented with 10% fetal bovine serum (Biochrom, Berlin, Germany), 2 mM L-glutamine, and 50 unit/ml penicillin/streptomycin (Sigma-Aldrich Chemie, Munich, Germany). Occasionally, previously described stable HEK293 cell lines expressing WT ClC4 or WT ClC-5 [19,28] were also used for electrophysiology; stable expression was maintained using 0.9 mg/ml Geneticin (G418) added to the standard HEK293 growth medium (MEM, supplemented with 10% FBS, both from Gibco, Life Technologies, Darmstadt, Germany).

### Electrophysiology

Whole-cell or inside-out patch-clamp measurements [59] were performed 48 h post-transfection using a HEKA EPC-10 amplifier and PATCHMASTER software (HEKA Electronics, Lambrecht, Germany), or an Axopatch 200B amplifier with Clampex software (Molecular Devices, Sunnyvale, CA, USA). Filtering with 3–10 kHz was applied prior to digitizing the data at a 30-kHz sampling rate. Patch pipettes from borosilicate glass were fabricated with a P-97 puller (Sutter Instrument, Novato, CA, USA) and heat-polished to reach a resistance between 0.9 and 2 MΩ on an MF-900 Microforge (Narishige, London, UK). If required (i.e. for inside-out measurements), pipettes were coated with dental wax to minimize their electric capacitance. Series resistance compensation and capacitive cancellation minimized the voltage error to < 5 mV. Linear capacitive artifacts were further reduced by a P/4 leak subtraction sequence [60] applied from a holding potential of −30 mV. The standard extracellular solution contained (in mM) 145 NaCl, 15 HEPES, 5 MgCl$_2$, 4 KCl, and 1 CaCl$_2$. The nucleotide-free pipette solution contained (in mM) 120 NaCl, 15 HEPES, 5 MgCl$_2$, and 1 EGTA. Adenine nucleotides (5 mM Na-ATP, 5 mM Na-ADP or 5 mM Na-AMP, all from Sigma-Aldrich) were added to the pipette solution shortly before the experiments and the pH was additionally adjusted. All solutions were titrated to pH 7.4 using NaOH. In NMDG-based solutions, NaCl was substituted by equimolar N-methyl-d-glucamine-Cl; pH was adjusted with NMDG.

### Gating current measurements to estimate the apparent CLC gating charge

The gating charge mobilized during CLC activation was obtained by integrating the area under P/4 subtracted off-gating currents elicited by voltage jumps in the dynamic range of the investigated CLC transporter [27]. For further analysis, the gating charges were normalized either to the maximal gating charge at saturating voltages or to the steady state of the ionic current at +165 mV.

To describe the voltage dependence of the charge movements ($Q(V)$), the following standard Boltzmann function (Equation 1) was fitted to the data:

$$Q(V) = \frac{A}{1 + e^{\beta(V - V_{0.5})}} \tag{1}$$

with:

$$\beta = \left(\frac{ze_0}{k_B T}\right) \tag{2}$$

Here, $V_{0.5}$ and $z$ denote the voltage for half-maximal activation, and the apparent number of elementary charges $e_0$ displaced in the transmembrane electric field during the investigated voltage-dependent transition. The Boltzmann constant and the absolute temperature are indicated as $k_B$ and $T$, respectively. The apparent amplitude of the Boltzmann function is indicated as $A$.

### Structural modeling and data analysis

The UCSF Chimera [61] interface to MODELLER [62] was used to create a 3D protein homology model of the full-length ClC-5 amino

acid sequence based on the crystallized structures of the soluble CBS domain of ClC-5 (PDB ID: 2J9L [6]) and the transmembrane domains of CmCLC (PDB ID: 3ORG [5]). MD movies were prepared using the standard settings of the "Morph conformations" and "MD movie" functions of UCSF Chimera [61]. Experimental data were analyzed using a combination of FitMaster (HEKA) or Clampfit (Molecular devices), Excel (Microsoft), and Origin (OriginLab Corporation, Northampton MA, USA). Statistical significance was assessed using a two-sample *t*-test. All summary data are shown as mean ± SEM.

Expanded View for this article is available online.

## Acknowledgements
We thank Birgit Begemann, Petra Killian, and Toni Becher for technical assistance. This work was supported by the DFG Research Unit FOR 2795, FA 301/13-1.

## Author contributions
MG, KL, REG, CF, and AKA contributed to the design of the work. MG, KL, REG, and AKA contributed to acquisition and analysis of the data. AKA drafted the manuscript. MG, KL, REG, CF, and AKA revised the paper critically for important intellectual content.

## Conflict of interest
The authors declare that the research was conducted in the absence of any commercial or financial relationships that could be construed as a potential conflict of interest.

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
