## [Review Process File · EMBO Reports]

Metabolic Energy Sensing by Mammalian CLC Anion/Proton Exchangers

Matthias Grieschat, Raul E. Guzman, Katharina Langschwager, Christoph Fahlke, and Alexi K. Alekov

Review timeline:

Submission date:	4 February 2019
Editorial Decision:	23 May 2019
Revision received:	3 June 2019
Editorial Decision:	12 July 2019
Revision received:	17 November 2019
Editorial Decision:	23 December 2019
Revision received:	26 January 2020
Accepted:	29 January 2020

Transaction Report:

1st Editorial Decision

23 May 2019

Thank you for the re-submission of your strengthened and revised manuscript to EMBO reports. Please accept my apologies for this very unusual delay in handling your manuscript. We have now received the full set of referee reports that is copied below. Unfortunately, referee 1 was not available anymore and I asked another expert in the field to evaluate your study and your response to the referee concerns.

I am sorry to say that the evaluation of your manuscript is overall not positive. Reviewer 1 considers the data insufficient to support the main claims. I note that Referee 2 and 3 are more positive but also these referees point out some remaining concerns and are not completely convinced of the data and referee 2 suggests further experiments to support the main conclusions. With these important concerns remaining I am sorry to say that we cannot offer to publish your manuscript.

I am sorry to disappoint you on this occasion, and hope that the referee comments are helpful in your continued work in this area.

REFeree REPORTS

Referee #1:

The paper presents an analysis of the effects of Adenine nucleotides on CLC-3,4, and 5 transporter currents. I will not again summarize the details of the work, which was done in the first round of review. I agree with the previous reviewers that the analysis of rundown in whole-cell patches in an effort to understand the binding of nucleotides to the cytoplasmic domains of these proteins is not a sufficiently quantitative approach to analyze these data, and to do this properly would require inside-

out patch experiments. Nothing in the work demonstrates activation of the CLCs by nucleotide (the main claim of the manuscript), only changes in the rates of current decrease. I do not find the authors' responses to the previous reviews convincing.

Referee #2:

"Metabolic Energy Sensing by Mammalian CIC Anion/Proton Exchangers"

In this paper, the authors investigate the role of ATP, ADP, and AMP on three different CIC antiporters- CIC-3, CIC-4, and CIC-5. Using whole cell recordings, they find that ATP and ADP applied in the pipette protect the currents from rundown, suggesting that these nucleotides increase ion transport rates in these transporters. AMP, on the other hand, has the opposite effect and seems to slightly decrease transport rates. Interestingly, gating currents revealed that all 3 nucleotides tested shifted the voltage dependence of activation slightly towards more positive voltages, leading the authors to conclude that ATP and ADP enhance CIC ion transport by reducing the population of channels undergoing silent non-transporting cycles. In the case of CIC-5, they show the important controls- in an inside-out patch, the attenuation caused by ATP is immediate and reversible, and a mutation in the CBS domain nucleotide binding site abolishes the effect of ATP. Furthermore, they find that mutation of the internal gating glutamate but not the external gating glutamate alters the effects of ATP.

The authors have given careful consideration to reviewers' points from the previous submission and have addressed most of the issues. I find the paper very interesting but still have a few concerns.

1) The experiments with excised inside-out patches provide an important control shown in Figure S1. This experiment however is only done with CIC-5 and ATP addition and only data from a single patch is shown. Some statistics including the percentage increase in current induced by ATP in several patches should also be included. Likewise, it is important to show that the effects of ADP and AMP observed in Figure 1B are also consistent in the inside-out patch configuration. As there are differences reported between CIC-3, -4, and -5, ideally this control would have been done in all 3 transporters. If this was not possible due to low current levels, the authors should make some mention of this.

2) The discrepancy with the results of the Pusch lab has been explained as well as could be expected in my opinion. That said, I do not believe that the perfusion speed should have been an issue using large pipettes with patches in the inside-out configuration, as many people have observed ligand-induced changes in currents on the millisecond time scale while using large pipettes. I do agree that low currents, leak subtraction, and lack of clarity as to whether Mg^{2+} was also added could possibly explain the disparity. In this paper too, a P/4 leak subtraction protocol was used. If leak is changing throughout these 10-min experiments, this subtraction protocol could introduce artifacts as well, even though the currents are larger in the whole cell configuration. The authors should confirm that the effects that they observe are qualitatively seen in the absence of P/4 subtraction as well for this study.

3) The Y617A result shown in Figure S2 is crucial for demonstrating that the observed effects are likely CIC-5-specific and not due to endogenous channels. Was the equivalent mutation tested in CIC-3 or CIC-4?

Minor comments:

1) Figure 1A legend: "Some of the nucleotide-coordinating residues between in" should be grammatically corrected.

2) Figure 1B: This would be clearer if the authors colored the first and last trace differently. Also it is not clear the time between sweeps. As noted above, it should be noted if the same qualitative results are observed in the absence of P/4 subtraction.

3) Page 9: Results on the nucleotide chemistry are very intriguing and support specificity of the binding pocket. However, the last sentence implies that the differences of observed effects are directly due to differences in binding. It is possible that the phosphate group is necessary not for binding but for propagating the effect of a bound ligand, and this possibility should at least be mentioned.

4) Figure S1: The authors state in response to reviewers that they mention that SCN uncouples proton and chloride transport here, but this mention is not obvious still.

5) Figure S4A: Reference to Figure 3D must be corrected as there is no figure 3D.

Referee #3:

This revised manuscript by Grieschat and colleagues investigates how the CLC exchangers are modulated by adenosine nucleotides. I find this revised version improved; the excised patch experiments in Supp Fig. S1 are convincing and the finding that the gating-incompetent mutant E211G retains modulation by nucleotides is an important positive control and supports the authors' hypothesis that nucleotides modulate the probability that the transporter is in an 'activated' or 'non-transporting' conformations.

I still have some reservations as to how the authors explain the discrepancies between their findings and those of other groups. For example I am puzzled by the finding that AMP, ADP and ATP have different functional effects on CLC-5, while having nearly identical measured affinity for the C-ter of CLC-5 (Meyer et al., NSMB, 2007). I also do not think that the results from Wellhauser et al are indicative of an 'induced fit' mechanism. Similarly, I think that the authors' suggestion that the currents measured by Zifarelli and Pusch might be endogenous single channels is quite odd, since the currents reported in that manuscript are very obviously ensemble measurements and incompatible with those expected for a single channel.

Nonetheless, I think that the present manuscript adds an interesting twist to an important question. It is likely that further work will be needed to reconcile the differences between groups and clarify the mechanism of CLC modulation by nucleotides.

I would encourage the authors to tone down some of their statements. For example, they use words like "demonstrate" several times (in the abstract, results and discussion), where I think that terms such as "suggest" would be more appropriate. The measurements presented here are indirect, and the inferences drawn are shrouded by uncertainties due to our poor understanding of CLC exchanger function, making mechanistic interpretation of the data less definitive than a "demonstration" would imply.

1st Revision – authors' response

3 June 2019

Referee #1

"I agree with the previous reviewers that the analysis of rundown in whole-cell patches in an effort to understand the binding of nucleotides to the cytoplasmic domains of these proteins is not a sufficiently quantitative approach to analyze these data, and to do this properly would require inside-out patch experiments. Our goal was to evaluate the functional effects of intracellular nucleotides on the intracellular CLC transporters CLC-3, CLC-4, and CLC-5. To this end, we used whole-cell patch clamp, which is a well-established and widely used experimental method for investigations protein-mediated transmembrane conductances. Whole-cell patch clamp has been successfully used to investigate the functional effects of various intracellular factors, including Ca^{2+} , local anesthetics, adenine nucleotides, pH, etc. Specifically relevant for our work, whole-cell patch clamp investigations have been successfully used to quantitatively describe the functional effect of intracellular adenine nucleotides on different CLC proteins (e.g. Niemeyer et al., 2004, doi:10.1152/physiolgenomics.00070.2004; Bennetts et al., 2005, doi: 10.1074/jbc.M502890200; Bennetts et al., 2007, doi: 10.1074/jbc.M703259200; De Angeli et al., 2009, doi: 10.1074/jbc.M109.005132; Dave et al., 2010, doi: 10.4161/chan.4.4.12445; Bennetts et al., 2012, doi: 10.1074/jbc.M111.327551; Stölting et al., 2013, doi: 10.1007/s00424-013-1286-0; Yamada and Strange, 2016, doi: 10.1016/j.bpj.2016.09.037; Yamada et al., 2016, doi: 10.1016/j.bpj.2016.09.036). We would like to reiterate that the transporters investigated by us are intracellular proteins and that their plasma membrane expression is very limited. In addition, no specific chemical modulators of their function have been discovered yet (neither blockers nor activators). Therefore, whole-cell patch clamp has two important advantages over using inside-out measurements: much larger specific currents of the expressed transporters compared to the endogenous currents of the experimental system, and the possibility to quantitatively describe effects on voltage-dependent gating. The requirement that investigation of intracellular nucleotide effects should be conducted exclusively in the inside-out configuration appears, therefore, not justified.

“Nothing in the work demonstrates activation of the CLCs by nucleotide (the main claim of the manuscript), only changes in the rates of current decrease.”

The physiologically relevant finding of our study is that ATP, ADP, and AMP differentially regulate the intracellular transporters CIC-3, CIC-4, and CIC-5. To support our conclusion, we did not use the different current decrease rates but evaluated differences in the steady-state currents (Figure 1 in the manuscript), the differential regulation of voltage-dependent gating, and the differential effects on the gating-current/ionic-transport ratio (Figure 2 in the manuscript). In addition, we used inside out measurements to exclude that the effects of ATP might be due to altered endocytosis or exocytosis of CIC-5 (Figure S1 in the manuscript) and described the effects of ATP on mutant Y617A to demonstrate the specificity of the effects (Figure S2 in the manuscript). These findings are independent of the experimental configuration (whole-cell or inside-out patch clamp). We provide an additional figure that substantiates this conclusion (Figure R1 for the reviewers). Specifically, Figures R1A and R1B illustrate that the perfusion rates in our experiments are fast and that 10 minutes are sufficient for adequate cytoplasm-pipette exchange. We also provide additional analysis of the whole-cell capacitances and series resistances of the investigated CIC-3 cells (CIC-3 exhibits the largest difference in the decay rates). The cell capacitance provides information on the cell size and the series resistance – on how good the exchange between the pipette and the cytoplasm is. The similar values in the different data sets exclude the possibility that the observed effects are due to differences in the cell size or seal geometry (Figure R1C for the reviewers). Finally, Figure 1D illustrates the independence of the steady-state of a decaying process from its decay rate.

Referee #2:

“1) The experiments with excised inside-out patches provide an important control shown in Figure S1. This experiment however is only done with CIC-5 and ATP addition and only data from a single patch are shown. Some statistics including the percentage increase in current induced by ATP in several patches should also be included. Likewise, it is important to show that the effects of ADP and AMP observed in Figure 1B are also consistent in the inside-out patch configuration. As there are differences reported between CIC-3, -4, and -5, ideally this control would have been done in all 3 transporters. If this was not possible due to low current levels, the authors should make some mention of this.”

The CIC-5 current increase in the data presented in Figure S1 is 1.37 ± 0.06 (SEM). The included inside-out measurements were not intended as a control but as a test whether ATP depletion alters the exocytosis of CIC-5 as demonstrated for the endocytosis of CIC-2 (Dhani et al., 2008, PMID 17620322). To test the specificity of the effects in Figure 1, we investigated mutant Y617A CIC-5. In addition, we demonstrate differential effects of the nucleotides on voltage-dependent gating and the ratio between ionic transport and gating currents (Figure 2 in the manuscript). We would like to reiterate that the transporters investigated by us are intracellular proteins and that only a small number of the expressed proteins reside in the plasma membrane. This limits the experimentally measurable current amplitudes and renders excised patch measurements very unreliable. In addition, no specific chemical modulators of their function have been discovered yet (blockers or activators). Therefore, whole-cell patch clamp has two important advantages over using inside out patches: a) the specific currents of the expressed transporters are much larger than the endogenous currents of the experimental system, and b), the large currents allow a quantitative description of voltage-dependent gating.

To increase the CIC-5 currents in Figure S1, we used external SCN. However, SCN partially uncouples CLC transport. Therefore, such experiments are not directly comparable to the results in Figure 1. As requested, we could explicitly explain our choice of method. To better illustrate the problematic, we show a representative recording of CIC-4 in external Cl, first in the cell-attached mode and then after establishing the whole-cell configuration (Figure R2A). In addition, we show a recording of endogenous single channel conductances in untransfected cells. Please note their large amplitudes and long open times (Figure R2B). Finally, we would like to point out that whole-cell patch-clamp investigations have been successfully used to quantitatively describe the functional effect of intracellular adenine nucleotides on various CLC proteins (e.g. Niemeyer et al., 2004, doi:10.1152/physiolgenomics.00070.2004; Bennetts et al., 2005, doi: 10.1074/jbc.M502890200; Bennetts et al., 2007, doi: 10.1074/jbc.M703259200; De Angeli et al., 2009, doi: 10.1074/jbc.M109.005132; Dave et al., 2010, doi: 10.4161/chan.4.4.12445; Bennetts et al., 2012, doi: 10.1074/jbc.M111.327551; Stölting et al., 2013, doi: 10.1007/s00424-013-1286-0; Yamada and Strange, 2016, doi: 10.1016/j.bpj.2016.09.037; Yamada et al., 2016, doi: 10.1016/j.bpj.2016.09.036).

“That said, I do not believe that the perfusion speed should have been an issue using large pipettes with patches in the inside-out configuration, as many people have observed ligand-induced changes in currents on the millisecond time scale while using large pipettes. I do agree that low currents, leak subtraction, and lack of clarity as to whether Mg²⁺ was also added could possibly explain the disparity.”

As we do not have access to the original data of Zifarelli and Pusch, and as critical parts of the experimental procedures were not described, we can only hypothesize about the possible explanation that could have led to the effects observed by these authors. We mentioned the

perfusion because the membrane geometry of the excised patches can vary significantly depending on the sealing method. Slow application of low negative pressure might result in large membrane patches engulfed in the patch pipette. This method would increase the measured ionic currents and improve seal stability; however, it will also impair the solution exchange. In our manuscript, we have outlined endogenous conductances as the most probable explanation (in our opinion). Our conclusion is based on the different noise characteristics and the different kinetics of the current traces in the paper of Zifarelli and Pusch.

"In this paper too, a P/4 leak subtraction protocol was used. If leak is changing throughout these 10-min experiments, this subtraction protocol could introduce artifacts as well, even though the currents are larger in the whole cell configuration. The authors should confirm that the effects that they observe are qualitatively seen in the absence of P/4 subtraction as well for this study."

We are confident that the used P/4 leak subtraction did not affect our results. Specifically, we recorded the traces both with and without leak subtraction (See Figure R3). The feature is incorporated in the electrophysiology software (here HEKA) to allow controlling the recording quality. In Figure R3 we show data for CIC-3, the transporter that should be most prone to artifacts because it exhibited the smallest macroscopic currents (0.7±0.2 nA, SEM, n=20).

"3) The Y617A result shown in Figure S2 is crucial for demonstrating that the observed effects are likely CIC-5-specific and not due to endogenous channels. Was the equivalent mutation tested in CIC-3 or CIC-4?"

These experiments have not been required during the previous revision and we did not test the corresponding mutants. However, we are confident that we can conduct these experiments in a reasonable time period. We would, however, put a note of cautiousness: Y617A leads to defective CIC protein glycosylation (Wellhauser et al., 2010, DOI 10.1074/jbc.M110.175877). Therefore, the surface expression of the CIC-3 and the CIC-4 mutants might be insufficient.

Minor comments:

"1) Figure 1A legend: "Some of the nucleotide-coordinating residues between in" should be grammatically corrected."

We apologize for the mistake. Should read: "Some of the nucleotide-coordinating residues".

"2) Figure 1B: This would be clearer if the authors colored the first and last trace differently. Also it is not clear the time between sweeps."

We thank the Referee for the suggestion and will take the advice. The time between the sweeps is 10s, however, some of the traces were not shown to improve the visibility of the individual sweeps.

"3) Page 9: Results on the nucleotide chemistry are very intriguing and support specificity of the binding pocket. However, the last sentence implies that the differences of observed effects are directly due to differences in binding. It is possible that the phosphate group is necessary not for binding but for propagating the effect of a bound ligand, and this possibility should at least be mentioned."

We thank the Referee for the suggestion and will take the advice.

"4) Figure S1: The authors state in response to reviewers that they mention that SCN uncouples proton and chloride transport here, but this mention is not obvious still."

We agree and will include this in the manuscript.

"5) Figure S4A: Reference to Figure 3D must be corrected as there is no figure 3D."

We apologize for the mistake.

Referee #3:

"I still have some reservations as to how the authors explain the discrepancies between their findings and those of other groups. For example, I am puzzled by the finding that AMP, ADP and ATP have different functional effects on CLC-5, while having nearly identical measured affinity for the C-ter of CLC-5 (Meyer et al., NSMB, 2007)."

This is a very interesting question that we cannot answer at this time. However, we would like to reiterate that the CBS domains of mammalian AMPK proteins are also characterized by their almost identical binding affinities towards these three nucleotides. (Xiao et al., 2007, 2011) and that the AMPK activity is also differentially regulated by intracellular nucleotides. Please, note also our response to the next comment.

"I also do not think that the results from Wellhauser et al are indicative of an 'induced fit' mechanism."

We agree with the Referee. The results from Wellhauser et al. indicate that the C-termini undergo a significant conformational change upon ATP binding. We hypothesized that such conformational changes might affect also the binding pocket and alter nucleotide binding. However, the published structures (Meyer et al., NSMB, 2007) also indicate that the geometry of the binding pocket changes upon nucleotide binding. We have prepared two movies (best viewed when put on loop) produced by morphing the structure PDB 2ja3 (CIC-5, ADP-bound) into 2j9l (CIC-5, ATP-bound), both from the above paper. For clarity, only the ATP molecule from 2j9l has been shown. The first movie (cterm.mp4) shows that the conformation of the crystalized dimeric C-terminus depends on the bound nucleotide. Please note that the observed conformational changes support also our hypothesis that the movement is propagated towards the protein core positioned above the C-termini (the

“loose” ends of the structures). The second movie (cterm2.mp4) depicts enlarged the nucleotide binding pocket and specifically the movements of amino acid Y617.

“Similarly, I think that the authors' suggestion that the currents measured by Zifarelli and Pusch might be endogenous single channels is quite odd, since the currents reported in that manuscript are very obviously ensemble measurements and incompatible with those expected for a single channel.”

As we do not have access to the original data of Zifarelli and Pusch, and as critical parts of the experimental procedures were not described, we can only hypothesize about the possible explanation that could have led to the effects observed by these authors. In our manuscript, we have outlined the most probable reason (in our opinion). Our conclusion is based on the different noise characteristics (high-frequency regular noise vs. low-frequency irregular noise) and the different kinetics of the current traces in the paper of Zifarelli and Pusch. Please note the different duration of the traces in the manuscript of Zifarelli and Pusch. Especially the different noise characteristics suggest that some uncharacteristic conductances might have been added or subtracted. Please note also the long opening times of the single channels presented by us in the figure for the Reviewers R2. Such openings could easily alter the measured current amplitudes without obviously changing the macroscopic current characteristics. Nevertheless, we would point at the necessity of further investigations.

“I would encourage the authors to tone down some of their statements. For example, they use words like "demonstrate" several times (in the abstract, results and discussion), where I think that terms such as "suggest" would be more appropriate. The measurements presented here are indirect, and the inferences drawn are shrouded by uncertainties due to our poor understanding of CLC exchanger function, making mechanistic interpretation of the data less definitive than a "demonstration" would imply.”

We agree with the Referee. We will take the advice and tone down some of our statements, especially regarding the mechanistic interpretations of our results.

Figure R1

Figure R1. **A)** Cytoplasm-pipette exchange visualized by the fluorescence increase after obtaining the whole-cell configuration. The pipette solution contained 100 μ M BCECF. Fluorescence was excited by illumination at 480 nm and 440 nm and detected by a stabilized PMT. **B)** Time course of the fluorescence increase. The red line indicates an exponential fit with a time constant $t=77$ s. Please note that the molecular weight of BCECF (881g/mol) exceeds the one of ATP (507g/mol). Therefore, ATP perfusion will be even faster (Pusch and Neher 1998, PMID 2451806). **C)** Series resistances and cell capacitances of the cells used in the CIC-3 experiments (Fig. 1 in the manuscript) as a measure for the pipette diameter and cell size, respectively. **D)** The time constant does not influence the steady-state value of a decaying process. A standard exponential decay was calculated for five different time constants (the function is indicated in the figure).

Figure R2

voltage range: -115 mV to +165 mV

Figure R2. **A)** Ionic currents recorded from a cell transfected with CIC-4, first in the cell-attached mode (upper panel), and after establishing the whole-cell configuration (lower panel). The seal resistance in the cell-attached mode exceeded 10 G Ω , the patch pipette resistance was 0.6 K Ω , the measurements were conducted in Cl-based solutions. **B)** Endogenous single channels in untransfected TSA cells, 10kHz filter. Currents were elicited by voltage steps to -100mV. The patch pipette resistance was 10 M Ω , experiments were conducted in Cl-based solutions.

Figure R3

Figure R3. A), B) Representative current traces from a cell expressing CIC-3 recorded without (a) and with (b) leak subtraction (same recording, displayed in two different modes). C) Effects of adenine nucleotides or their washout on CIC-3 ion transport (n=5 in each data set, total 20). The current amplitudes were measured at the end of the test pulses using data as shown in (A, B), and normalized to the initial current amplitudes obtained after establishing the whole-cell configuration D), E) Series resistances and cell capacitances of the cells used in the CIC-3 experiments (Fig. 1 in the manuscript) as a measure for the pipette diameter and cell size, respectively.

Thank you once more for your message asking us to reconsider our decision and invite revision of your manuscript. I have meanwhile received feedback from referee 2 on your point-by-point response and we have also earlier received feedback from referee 3.

Referee 2 is overall satisfied with your response and does not oppose publication. S/he notes that "... there are issues pointed out by all of the reviewers, but I also understand the challenges of these experiments and do feel that the authors have done their best to control for what they can overall". This referee still considers info on CLC-3 and CLC-4 mutants a valuable addition, but I suggest to include these experiments only if they can be accomplished in a reasonably short time frame. Referee 3 noted also that despite some concerns like the lack of quantification (as also pointed out by referee 1), "[...] I do think that the data in the manuscript makes a reasonable case that the different nucleotides have different regulatory effects on transporter activity". This referee supports publication if some of the statements regarding the mechanistic interpretation are toned down.

The two referees that had also seen the first version of your manuscript are thus supportive of publication if the remaining concerns are addressed, potential limitations are discussed and statements on the mechanism are toned down. I would therefore ask you to revise your manuscript along the lines suggested by the referees.

From the editorial side there are also a number of things that we need:

- Please note the nomenclature "Appendix" for the Supplementary information. Please follow the nomenclature Appendix Figure Sx and Appendix Table Sx throughout the text and also label the figures and tables according to this nomenclature. Please also include a table of content on the first page of the Appendix pdf that also gives page numbers. For more details please refer to our guide to authors.

- Movies should be uploaded as a .zip file that contains the movie itself and its legend in the format of a README.txt file.

- Please note that the abstract should present all findings in present tense.

- You have currently 4 figures and your manuscript is scheduled to be published as Report. In this case the Results & Discussion section should be combined. If however you feel that an extended Discussion is essential, please contact me so that we can discuss whether the manuscript could be published in our Article section.

- Figure legends: please ensure to add a description of the bars and error bars (sd, sem), the number of samples analysed, the test used to calculate the p-values, and a definition of the p-values displayed to all panels displaying quantitative data.

This is currently missing/incomplete for Fig. 1D, Fig. 2B, C, Fig. 3B, Fig. 4, Fig. S2, S3, S4, S5C, S6,

- Please provide a complete author checklist, which you can download from our author guidelines (<<https://www.embopress.org/page/journal/14693178/authorguide>>). Please insert information in the checklist that is also reflected in the manuscript. The completed author checklist will also be part of the RPF.

- Finally, EMBO reports papers are accompanied online by A) a short (1-2 sentences) summary of the findings and their significance, B) 2-3 bullet points highlighting key results and C) a synopsis image that is 550x200-400 pixels large (width x height). You can either show a model or key data in the synopsis image. Please note that the size is rather small and that text needs to be readable at the final size. Please send us this information along with the revised manuscript.

2nd Revision - authors' response

17 November 2019

Responding to the specific comments of Referee 2:

1. We point out the low current amplitudes in our inside-out experiments (page 5, lines 77-82 and page 6 line 85-86).
2. We include a figure comparing leak-subtracted and non-leak-subtracted data (Appendix Figure S1).
3. We succeeded in expressing the CIC-4 mutant analogous to the Y617A CIC-5. The data are included in Appendix Figure S3.
4. We have corrected the Figure 1B legend. We have colored the first and the last traces in Figure 1B, we mention the possibility that the phosphate group might be involved in propagating the conformational changes (page 11, line 213), we point in the manuscript to the uncoupling action of SCN (page p6 line 79-82), and we have corrected the wrong reference in figure S4A (now S5A).

Responding to the specific comments of Referee 2:

1. We have extended the discussion the discrepancies between our data and the toned down several of our statements: Abstract: lines 7-9, line 12; Introduction: page 4, line 51 (p4 line 51); Results and discussion: p5 line 70-74, p7 line 114-119, p8 line 141, p8 line 142, p8 line 146-149, p9 line 155, p10 line 191.
2. We discuss the limitation of our study concerning the quantification of the current amplitude decay times: p7 lines 106-110.

3rd Editorial Decision

23 December 2019

Thank you for your patience while we have reviewed your revised manuscript. I am now writing with an 'accept in principle' decision, which means that I will be happy to accept your manuscript for publication once a few very minor issues/corrections have been addressed.